DATA RELEASE

# The genome assembly and annotation of the Chinese cobra, *Naja atra*

Jiangang Wang[1,2,†], Yuxin Wu[1,†], Shiqing Wang[2,3], Weiwu Mu[1], Wenmei Zeng[1], Xi Chen[1], Kangfeng Jiang[1], Liangyu Yang[1], Guohai Hu[4,*] and Fengping He[1,*]

1 College of Veterinary Medicine, Yunnan Agricultural University, Kunming, 650231, China
2 State Key Laboratory of Agricultural Genomics, BGI-Shenzhen, Shenzhen, 518083, China
3 College of Life Sciences, University of Chinese Academy of Sciences, Beijing, 100049, China
4 China National GeneBank, BGI-Shenzhen, Shenzhen, 518120, China

## ABSTRACT

In China, 65 types of venomous snakes exist, with the Chinese Cobra *Naja atra* being prominent and a major cause of snakebites in humans. Furthermore, *N. atra* is a protected animal in some areas, as it has been listed as vulnerable by the International Union for Conservation of Nature. Recently, due to the medical value of snake venoms, venomics has experienced growing research interest. In particular, genomic resources are crucial for understanding the molecular mechanisms of venom production. Here, we report a highly continuous genome assembly of *N. atra*, based on a snake sample from Huangshan, Anhui, China. The size of this genome is 1.67 Gb, while its repeat content constitutes 37.8% of the genome. A total of 26,432 functional genes were annotated. This data provides an essential resource for studying venom production in *N. atra*. It may also provide guidance for the protection of this species.

**Subjects** Genetics and Genomics, Zoology, Animal Genetics

**Submitted:** 18 May 2023

\* Corresponding authors. E-mail: huguohai@cngb.org; hefengping@outlook.com

† Contributed equally.

Preprint submitted at https://doi.org/10.20944/preprints202311.0597.v1

Included in the series: *Snake Genomes* (https://doi.org/10.46471/GIGABYTE_SERIES_0004)

## INTRODUCTION

Elapidae is a family of snakes divided into three subfamilies (Bungarinae, Elapinae and Notechinae), with 44 genera and around 186 described species distributed widely [1]. The front of the mouth of an elapid has permanently erect tusks, which are his distinguishing features. Elapids include terrestrial and sea snakes. Terrestrial elapids, a family of venomous snakes, are distributed across the globe in tropical and subtropical regions, with most species inhabiting the Southern Hemisphere. Elapid sea snakes are mainly distributed in the Indian Ocean and the Southwest Pacific Ocean [2].

The Chinese cobra, or *Naja atra* (NCBI: txid8656) (Figure 1), is a species of cobra from the family Elapidae. Chinese cobras are usually between 1.2 and 1.5 m long [3], and they are among the most prevalent cobra species in China. The Chinese cobra likes to inhibit plains, hills and low mountains [4]. Humans often encounter Chinese cobras, although these snakes usually escape to avoid confrontation with humans. Chinese cobras can be observed hunting during daylight hours from March to October and up to 2–3 hours after sunset at temperatures of 20–32 °C [5]. They have a widely varied diet and prey on rodents, frogs, toads and other snakes.

The Chinese cobra is highly poisonous, its venom consisting mainly of postsynaptic neurotoxins and cardiotoxins [6]. Their venom offers them protection from predation to a

**Figure 1.** The view of the head of a Chinese cobra (*N. atra*) snake on alert in Tainan City. *N. atra.* Source: Boris Smokrovic, Unsplash, CC0

certain extent; however, populations of Chinese cobra have declined by 30% to 50% due to habitat loss and hunting by humans. The venom of Chinese cobras can be used to extract anti-cobra snake venom, which is used to treat cobra snake bites. Although the Chinese cobra is currently listed as a Vulnerable species on the International Union for Conservation of Nature Red List [7], its numbers in the wild have declined from Vulnerable to Endangered due to continued hunting.

## MAIN CONTENT

### Context

Snakebite is a serious threat to human life as it kills around 100,000 people annually. Genome-enabled research of toxin genes may facilitate the development of effective antivenoms. Here, we present a highly continuous reference genome assembly of *N. atra*. While there is a reference genome for the Indian cobra (*Naja naja*) [8], this is the first for the Chinese cobra. This resource may also provide valuable information for the conservation of this vulnerable species, which can be used for targeted protection and breeding.

## METHODS

The detailed methods used in this study are available via a protocol collection hosted in protocols.io [9] (Figure 2).

### Sample collection and sequencing

The *N. atra* sample used in this study was captured in Huangshan, Anhui, China, in 2021. After collection, the specimen was quickly frozen to −80 °C using drikold dry ice for storage and transport. Methods for DNA extraction, library construction and sequencing were identical those used by Liu *et al.* in a previous study [10].

Sample collection, experiments and research design were all authorized by the Institutional Review Board of BGI (BGI-IRB E22001). In this research, all the procedures have been operated abiding to the guidelines from BGI-IRB strictly.

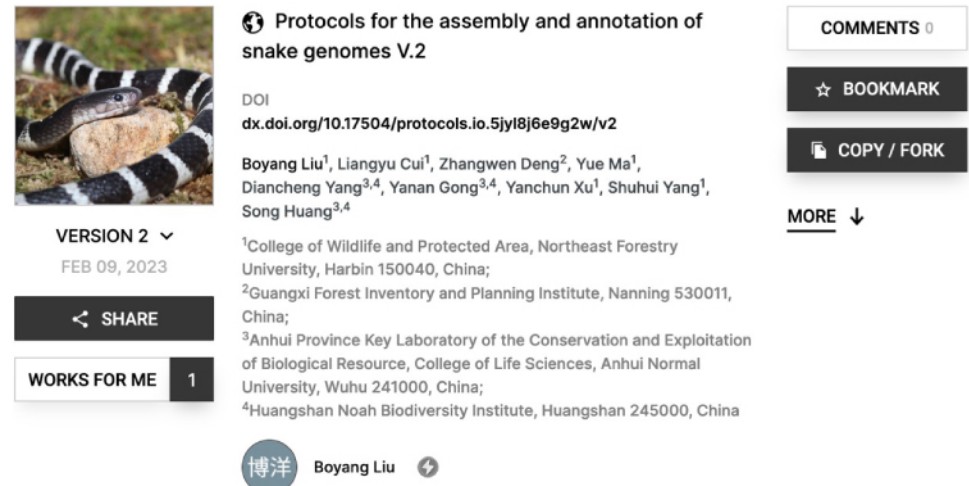

Protocols for the assembly and annotation of snake genomes V.2

DOI
dx.doi.org/10.17504/protocols.io.5jyl8j6e9g2w/v2

Boyang Liu[1], Liangyu Cui[1], Zhangwen Deng[2], Yue Ma[1], Diancheng Yang[3,4], Yanan Gong[3,4], Yanchun Xu[1], Shuhui Yang[1], Song Huang[3,4]

[1]College of Wildlife and Protected Area, Northeast Forestry University, Harbin 150040, China;
[2]Guangxi Forest Inventory and Planning Institute, Nanning 530011, China;
[3]Anhui Province Key Laboratory of the Conservation and Exploitation of Biological Resource, College of Life Sciences, Anhui Normal University, Wuhu 241000, China;
[4]Huangshan Noah Biodiversity Institute, Huangshan 245000, China

博洋  Boyang Liu

**Figure 2.** A protocols.io collection of the standard protocols for sequencing snake genomes [9].
https://www.protocols.io/widgets/doi?uri=dx.doi.org/10.17504/protocols.io.5jyl8j6e9g2w/v2

## Genome survey, assembly, annotation and assessment

The single-tube long fragment read sequencing data were assembled using Supernova (v2.1.1, RRID:SCR_016756) [11]. NextPolish (v1.0.5) [12] was then used to perform a second round of correction and a third round of polishing of this assembly using the Whole Genome Sequencing data. To get a haploid representation of the genome, duplicates were purged from the genome using the purge_dups pipeline (RRID:SCR_021173) [13]. The completeness of the genome was evaluated using sets of BUSCO (v5.2.2, RRID:SCR_015008) [14] with genome mode and lineage data from vertebrata_odb10 [15].

In order to detect the presence of known repeat elements in the genome of the many-banded *P. mucosa*, the following approach was employed. To identify the known repetitive elements in the genome of the many-banded krait, we used Tandem repeats Finder [16], LTR_Finder (RRID:SCR_015247) [17] and RepeatModeler (v2.0.1, RRID:SCR_015027) [18]. RepeatMasker (v3.3.0, RRID:SCR_012954) [19] and RepeatProteinMask v3.3.0 [20] were used to search the genome sequences for known repeat elements. The BRAKER2 pipeline (RRID:SCR_018964) [21] was used for gene prediction. Then, the gene sets were aligned against several known databases, including SwissProt, TrEMBL [22], Kyoto encyclopedia of genes and genomes (KEGG) [23], gene ontology (GO), and the Non-Redundant Protein Sequence Database [24] database.

## RESULTS

We present a draft genome sequence of *N. atra*. The size of this genome is 1.67 Gb (Table 1), similar to the previously published 1.79 Gb genome of *N. naja* [8]. The scaffold N50 length is 234.17 Kb, and the CG content reached 37.8%. The maximal scaffold length is 2,929,773 bp, demonstrating that the reference is highly continuous according to the characteristics of the genome sequence. In addition, the integrity of the genome was assessed at 84.1% using BUSCO (Figure 3).

In our *N. atra* genome, the content of repetitive elements is up to 40.26%, and the total length is 672 Mb (Tables 2, 3). After we counted all repeat elements, we found that long



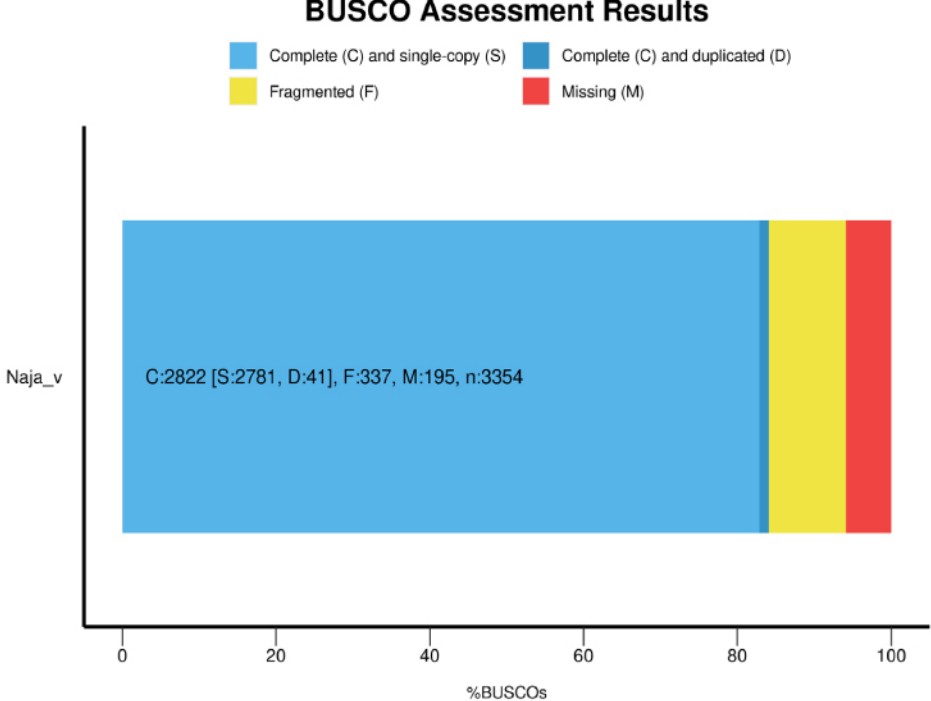

**BUSCO Assessment Results**

Complete (C) and single-copy (S)   Complete (C) and duplicated (D)
Fragmented (F)   Missing (M)

Naja_v    C:2822 [S:2781, D:41], F:337, M:195, n:3354

%BUSCOs

**Figure 3.** BUSCO assessment result of our *N. atra* genome.

**Table 1.** Summary of the features of our *N. atra* genome.

|  | Contig | Scaffold |
|---|---|---|
| Maximal length (bp) | 271,789 | 2,929,773 |
| N90 (bp) | 4,371 | 7,368 |
| N50 (bp) | 33,081 | 234,173 |
| Number ≥ 100 bp | 194,909 | 106,418 |
| Number ≥ 2 kb | 113,570 | 54,157 |
| GC content (%) | 40.3 | 37.8 |
| Genome size (bp) | 1,671,178,062 | |

**Table 2.** Statistics for repetitive sequences identified in our *N. atra genome*.

| Type | Length (bp) | % in genome |
|---|---|---|
| DNA | 37,917,702 | 2.269170 |
| LINE | 449,338,074 | 26.890460 |
| SINE | 2,779,035 | 0.166310 |
| LTR | 224,765,038 | 13.450975 |
| Other | 0 | 0 |
| Satellite | 632,498 | 0.037852 |
| Simple_repeat | 5,080,994 | 0.304070 |
| Unknown | 7,924,824 | 0.474258 |
| Total | 672,795,525 | 40.263183 |

interspersed nuclear elements (LINEs) accounted for 30.63%, long terminal repeats (LTRs) accounted for 14.03% and DNA accounted for 4.27% (Figure 4).

Finally, 29,063 functional genes were annotated. Through KEGG annotation, we found that the genes related to signal transduction are essential in *N. atra* (Figure 5).

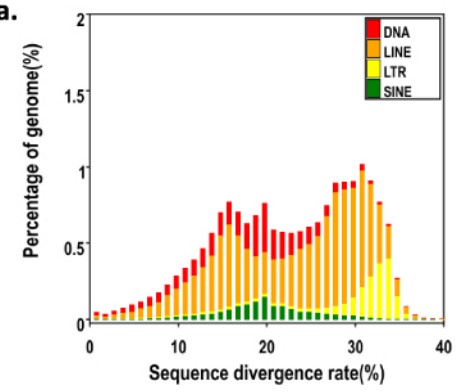
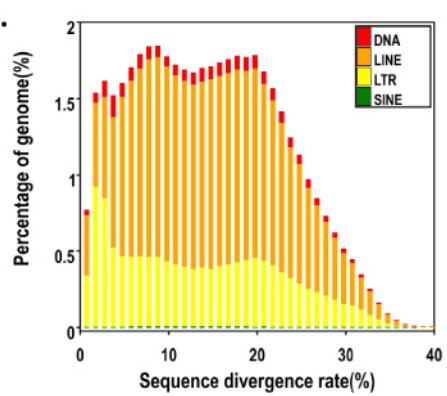

**Figure 4.** Distribution of transposable elements (TEs) in our *N. atra* genome. The TEs include DNA transposons (DNA) and RNA transposons (i.e., DNAs, LINEs, LTRs, and short interspersed nuclear elements (SINEs)). (a) Known sequences divergence rate (b) *De novo* sequences divergence rate.

**Table 3.** Summary of the TEs in our *N. atra* genome.

| Type | Repbase TEs | | TE proteins | | *De novo* | | Combined TEs | |
|---|---|---|---|---|---|---|---|---|
| | Length (bp) | % in genome | Length (bp) | % in genome | Length (bp) | % in genome | Length (bp) | % in genome |
| DNA | 44,907,141 | 2.57 | 3,638,477 | 0.20 | 41,761,899 | 2.39 | 81,259,555 | 4.66 |
| LINE | 170,663,721 | 9.79 | 140,023,530 | 8.03 | 581,624,764 | 33.36 | 619,156,475 | 35.51 |
| SINE | 25,759,131 | 1.47 | 0 | 0 | 8,061,060 | 0.46 | 32,226,226 | 1.84 |
| LTR | 22,468,876 | 1.28 | 30,088,483 | 1.72 | 149,994,747 | 8.60 | 159,624,403 | 9.15 |
| Other | 23,680 | 0.001 | 0 | 0 | 0 | 0 | 23,680 | 0.001 |
| Unknown | 0 | 0 | 0 | 0 | 5,653,213 | 0.32 | 5,653,213 | 0.32 |
| Total | 251,569,212 | 14.43 | 173,669,200 | 9.96 | 722,435,038 | 41.44 | 752,340,302 | 43.15 |

Furthermore, through a pathway enrichment analysis, we found that the number of Human Diseases pathways is the highest. Environmental Information Processing and Organismal systems also account for a relatively large proportion. According to the annotation and enrichment in the GO database, 6,292 genes are enriched in cellular process and 6,734 in binding.

## DATA AVAILABILITY

The data supporting the findings of this study have been deposited into the CNGB Sequence Archive (or CNSA) of China National GeneBank DataBase (or CNGBdb) with the accession number CNP0004141. Raw reads are available in the SRA via bioproject PRJNA955401. Additional data is in the GigaDB repository [25].

## EDITOR'S NOTE

This paper is part of a series of Data Release papers presenting the genomes of different snake species [26].

## ABBREVIATIONS

GO, gene ontology; KEGG, Kyoto encyclopedia of genes and genomes; LINE, long interspersed nuclear element; LTR, long terminal repeat; SINE, short interspersed nuclear element; TE, transposable element.

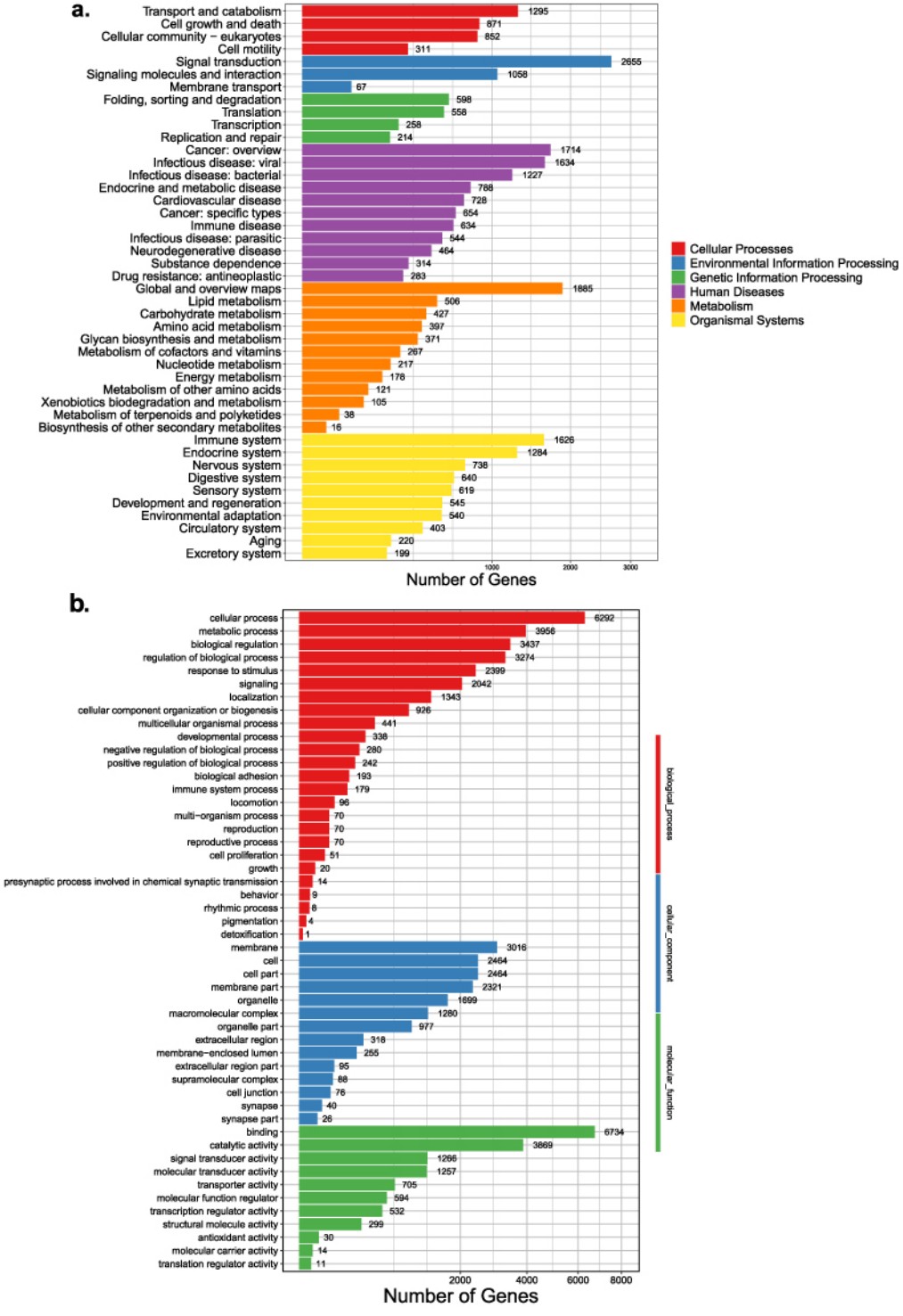

**Figure 5.** Gene annotation information of *N. atra*. (a) KEGG enrichment of *N. atra*. (b) GO enrichment of *N. atra*.

## DECLARATIONS

### Ethics approval and consent to participate

The authors declare that ethical approval was not required for this type of research.

### Competing interests

The authors declare no conflict of financial interests.

### Authors' contributions

GH and FH designed and initiated the project. YW performed DNA extraction, library construction and data analysis. JW wrote the manuscript. All authors read and approved the final manuscript.

### Funding

Our project was financially supported by the Guangdong Provincial Key Laboratory of Genome Read and Write (grant no. 2017B030301011). This work was also supported by China National GeneBank (CNGB).

### Acknowledgements

Yunnan Agricultural University collected the samples.

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
