## [Reviewer Report]

Comments on revised manuscriptThe authors have improved their manuscript a little bit. The paper is now more readable. Although I think the paper does not reach my standard, the new cobra genome data presented here is a contribution to the herp community. Since GigaByte focuses on less-complex, stand-alone datasets, the paper may be acceptable.

---

## [Editor Report]

Editor’s AssessmentThe Chinese cobra Naja atra is a highly venomous snake among the most prevalent cobra species in China. To help better understand the evolution and venom of cobra species, a 1.67Gb reference genome was sequenced, assembled and described in this work. During review some inconsistencies the data quality were fixed. With other cobra species already published, this data can be combined with these and other upcoming snake genome data to construct the evolutionary history of snakes and other reptiles as well as the genetic basis of venom.

---

## [Reviewer Report]

Reviewer name and names of any other individual's who aided in reviewer Kushal SuryamohanDo you understand and agree to our policy of having open and named reviews, and having your review included with the published papers. (If no, please inform the editor that you cannot review this manuscript.)YesIs the language of sufficient quality?YesPlease add additional comments on language quality to clarify if needed
Are all data available and do they match the descriptions in the paper? NoAdditional CommentsI was not able to access all the data - but I assume its available under the accession they provide in the paper - also will be nice to have the gene annotation table as part of a supplementary section in the manuscriptAre the data and metadata consistent with relevant minimum information or reporting standards? See GigaDB checklists for examples <a href="http://gigadb.org/site/guide" target="_blank">http://gigadb.org/site/guide</a>YesAdditional CommentsIs the data acquisition clear, complete and methodologically sound?NoAdditional CommentsA generic protocol is referenced - I suggest the data collected specific to this snake be stated in the MSIs there sufficient detail in the methods and data-processing steps to allow reproduction?YesAdditional CommentsIs there sufficient data validation and statistical analyses of data quality? Not my area of expertiseAdditional CommentsThere is no statistical analysis applicable in the workIs the validation suitable for this type of data?NoAdditional Commentsnot sure this is applicable Is there sufficient information for others to reuse this dataset or integrate it with other data?NoAdditional Commentsnot clear - may be it is - urge the authors to make the gene annotation available as part of the manuscriptAny Additional Overall Comments to the AuthorThe authors report the assembly for Naja atra. The report a genome of 1.34Gb coding for 26,432 functional genes. The work is suitable for the journal. Some suggestions to improve the work to make it more than just a genome announcement paper are noted below.  1. Authors point to a protocols paper to figure out the data generate – will be useful for them to state what data type and much was collected 2. Suggest adding numbers that indicate how many scaffolds greater than 5 or 10Mb is present in the assembly and does that number roughly correspond to the number of chromosomes seen elapids  3. They say they identified 26,432 functional genes – is all of these full length protein coding genes ? Is there an annotation for toxin genes ? Can they state how many toxins they find in the genome given it’s a medically important snake as stated in their introduction.
RecommendationMinor Revision

---

## [Reviewer Report]

Reviewer name and names of any other individual's who aided in reviewer Peng ZhangDo you understand and agree to our policy of having open and named reviews, and having your review included with the published papers. (If no, please inform the editor that you cannot review this manuscript.)YesIs the language of sufficient quality?NoPlease add additional comments on language quality to clarify if needed
Are all data available and do they match the descriptions in the paper? NoAdditional CommentsAre the data and metadata consistent with relevant minimum information or reporting standards? See GigaDB checklists for examples <a href="http://gigadb.org/site/guide" target="_blank">http://gigadb.org/site/guide</a>NoAdditional CommentsIs the data acquisition clear, complete and methodologically sound?NoAdditional CommentsIs there sufficient detail in the methods and data-processing steps to allow reproduction?NoAdditional CommentsIs there sufficient data validation and statistical analyses of data quality? NoAdditional CommentsIs the validation suitable for this type of data?NoAdditional CommentsIs there sufficient information for others to reuse this dataset or integrate it with other data?NoAdditional CommentsAny Additional Overall Comments to the AuthorThe article does not meet the requirements of the Gigabyte Journal due to the following reasons, which require improvements:  The English writing is of poor quality and the expressions used are colloquial.  The data presented in the article is inconsistent. For instance, the genome size is mentioned as 1.34G in the abstract, but it is stated as 1.67G in the results section. This inconsistency raises concerns among reviewers regarding the reliability of the article.  The quality of the genome is very low, indicated by a high number of contigs and a low Busco score, suggesting that the genome is incomplete. The study appears to be unfinished with respect to the N.atra genome.  The previously published cobra genome, N.naja, exhibits higher quality. It possesses a contig N50 of 302.5 kb and represents a chromosome-level genome.  The usage of screenshots as primary figures is inappropriate, and the figure legends are too brief. I recommend using the experimental subject as Figure 1 instead.  The methods section lacks a description of the annotation method employed, requiring additional details.  In summary, my suggestion is to reject the article.RecommendationReject (Unsound or Unusuable)